# Intra-Vaginal Bio-Stimulation and Clitoral Massage to Enhance Pregnancy Rates in Water Buffalo in Coastal Bangladesh [note 1]

**DOI:** 10.3390/ani15040597

**Published:** 2025-02-19

**Authors:** Ashit Kumar Paul, Md. Fakruzzaman, Md. Ashadul Alam, Gautam Kumar Deb, M. A. M. Yahia Khandoker

**Affiliations:** 1Theriogenology and Reproductive Biotechnology Lab., Department of Medicine, Surgery and Obstetrics, Faculty of Animal Science and Veterinary Medicine, Patuakhali Science and Technology University, Outer Campus, Barishal 8210, Bangladesh; 2Department of Genetics and Animal Breeding, Faculty of Animal Science and Veterinary Medicine, Patuakhali Science and Technology University, Outer Campus, Barishal 8210, Bangladesh; 3Buffalo Research and Development Project, Bangladesh Livestock Research Institute, Savar, Dhaka 1341, Bangladesh; ashadul@blri.gov.bd (M.A.A.); debgk2003@blri.gov.bd (G.K.D.); 4Department of Animal Breeding and Genetics, Faculty of Animal Husbandry, Bangladesh Agricultural University, Mymensingh 2202, Bangladesh; yahiakhabg@bau.edu.bd

**Keywords:** *Bubalus bubalus*, reproductive management, coastal livestock farming

## Abstract

The pregnancy rate in buffalo cows after artificial insemination (AI) is lower than that of natural services. The penile sensation during mating is important for successful fertilization. Penile intra-vaginal bio-stimulation is absent in AI, which significantly decreases the pregnancy rate. Therefore, the objective of this study is to increase the pregnancy rate through intra-vaginal bio-stimulation with a penis-like device or artificial bull penis. It was determined that the application of an artificial penis during AI increased the pregnancy rate (42.2%) to the only perform AI group (32.5%) and clitoral massage group (37.5%). It was found that the intra-vaginal bio-stimulation subsequent to AI with both artificial penis and clitoral massage dramatically increased the pregnancy rate (52.5%) of buffalo cows to that of only the conduct AI group (32.5%). It is concluded that the use of an artificial bull penis as well as massage of the clitoris increased the pregnancy rate of buffalo cows.

## 1. Introduction

Water buffalo (*Bubalus bubalis*) are a crucial livestock species, contributing significantly to the agricultural economy of Bangladesh, particularly in coastal regions. The number of buffalo inhabitants in Bangladesh is 1.471 million, which are mostly reared in the coastal region as *bathan* practice (free range rearing of animals at fallow land and river-basin area) [1]. Buffalo’s populaces in Bangladesh are typically the native type that covers swamp and riverine types, which are distributed all over the country. Nevertheless, about 40% of the entire buffalo are in the coastal area, Jamuna-Brahmaputra and Meghna-Ganga floodplains measured to be buffalo prone [2]. Coastal buffaloes are salinity tolerant and capable of roaming in saline water. While buffaloes exhibit relative resistance to infectious diseases, they are prone to various reproductive disorders [3]. Improving reproductive efficiency in water buffalo is a priority for researchers and farmers alike. Low availability of feed or low quality of forage required during reproductive events, inadequate estrus detection, and artificial insemination (AI) efficiency often limit successful breeding in buffalo. In this context, bio-stimulation methods, such as intra-vaginal stimulation and clitoral massage, have emerged as promising alternatives to enhance estrus expression, facilitate successful insemination, and ultimately increase pregnancy rates [4]. Bio-stimulation is a natural or mechanical process used to induce physiological responses that enhance reproductive performance. This technique leverages sensory stimuli to improve hormonal activity, ovarian function, and uterine receptivity, creating favorable conditions for successful conception [5]. Both natural mating and artificial insemination systems are prevalent in farming practices in Bangladesh, although artificial insemination adoption remains limited. This is largely due to challenges such as seasonal breeders, poor estrus signs, lower pregnancy rate, and variability of estrus length in buffaloes. Moreover, buffalo also tend to show heat signs primarily at night, which poses difficulties for farmers in observation. During the estrus cycle of buffalo cows, the farmers are looking for bull buffaloes instead of AI for breeding purposes. This preference is partly due to the interval of 30 h between standing estrus and ovulation in buffaloes, a critical factor for successful artificial insemination [6]. In natural breeding, the pregnancy rate is higher than in artificial insemination. It happens because the bull can recognize the suitable time of estrus of buffalo cows, the volume of the semen, the concentration of semen as well as the bio-stimulation of the penis during mating [4]. The genital tract, such as clitoris stimulation at artificial insemination, may influence the rate of pregnancy in cattle as explored in several studies and increase pregnancy rates by 6.3 to 7.5% in cows [5,7]. It is reported that the pregnancy rate in buffalo in the coastal region of Bangladesh in natural service was 64%, whereas it was 32% in AI [8]. Previously, a penis-like device (PLD) was used after AI for intra-vaginal sensation and got about 15.5% higher pregnancy rates in cows [4] and 12% higher in female buffaloes [8] when compared with only the AI technique. However, Intra-vaginal bio-stimulation involves the use of mechanical devices designed to mimic the natural stimulation provided by the bull’s penis during mating. These devices stimulate the vaginal walls and cervix, promoting the release of reproductive hormones such as oxytocin and luteinizing hormone (LH). Oxytocin facilitates uterine contractions and sperm transport, while LH induces ovulation, ensuring optimal timing for fertilization [4,8]. In this study, we used the modified penis-like device/artificial bull penis for intra-vaginal bio-stimulation immediately following artificial insemination in buffaloes. As far as our knowledge extends, there has been no prior investigation into the application of both intra-vaginal bio-stimulation with a mPLD and clitoral massage subsequent to artificial insemination in buffaloes. Therefore, the objective of this study is to evaluate the effect of intra-vaginal bio-stimulation using a modified penis-like device and clitoral massage on the pregnancy rates of water buffalo in the coastal region of Bangladesh.

## 2. Materials and Methods

The study was conducted with the ethical approval of the Institutional Ethical Committee, Patuakhali Science and Technology University (Ref. No. PSTU/IEC/2025/09). This study was carried out from July 2023 to June 2024 at the Theriogenology and Reproductive Biotechnology Laboratory under the Department of Medicine, Surgery and Obstetrics, Faculty of Animal Science and Veterinary Medicine, Patuakhali Science and Technology University, Bangladesh.

### 2.1. Materials

#### 2.1.1. Anthelmintics, Vitamins and Minerals

The broad spectrum anthelmintics in a combination of levamisole 600 mg and triclabendazole 900 mg {Renadex^®^ Vet 2 g bolus (Renata Animal Health, Dhaka, Bangladesh) @ 1 bolus/75 kg body weight and repeated after 7 days}, vitamin AD_3_E {Renasol AD_3_E^®^ Vet 30 mL injection (Renata Animal Health, Dhaka, Bangladesh) @ 10 mL intramuscularly 7 days interval for three injections} were purchased from Renata Animal Health, Dhaka, Bangladesh.

#### 2.1.2. Modified Penis-like Device (mPLD)/Artificial Bull Penis

The mPLD/artificial penis was made according to the procedure described by Biswas et al. [4]. It was modified according to the length of the buffalo bull penis. Figure 1, which describes the dimensions of the artificial bull penis. The measurement was 20 cm in length, the diameter ranged from 12 cm to 5 cm from the base to tip gradually, and the handle was 12 cm. In this device, there are two openings to pass the AI gun through the device during insemination. We made another opening upper the handle to push warm water into the device for a warm sensation like an erect penis buffalo bull.

#### 2.1.3. Semen

The semen straws were collected from the buffalo breeding station, Bangladesh Livestock Research Institute, Savar, Dhaka. In some contexts, semen from the government breeding station, Department of Livestock Services, Savar, Dhaka, was used. The semen straws were preserved in liquid nitrogen as the standard protocol of cryopreservation.

### 2.2. Methods

#### 2.2.1. Study Area and Period

The study was conducted in the selected coastal areas of Bangladesh, which were Charfasson upazila (sub-district) in Bhola district (Latitude: 22°11′4.92″ N, Longitude: 90°45′45.00″ E) and Bauphal Upazila in Patuakhali district (Latitude: 22°25′45.12″ N, Longitude: 90°30′50.04″ E). The laboratory work was conducted during the period from July 2023 to June 2024.

#### 2.2.2. Selection and Management of Buffalo Heifers/Cows

A total of 200 buffalo heifers/cows were selected using a simple random sample method from the study area for the experiment based on research communication, health status, calving interval, farmer’s accountability, etc. Different types of inspections were performed to select buffalo heifers/cows. Age, parity, body condition score, reproductive health status, and previous calving difficulties were recorded. The breed of buffalo in the study area was mostly non-descriptive indigenous types. Therefore, we did not consider the breed as an influencing factor in the study. During selection, anthelmintics and vitamin AD_3_E were administered. It was suggested to inform us when the buffalo heifer/cows come to estrus. Then, after observing the sign of estrus, experimental procedures were performed. Most of the buffaloes were reared in the *bathan* system. Farmers were bringing their animals for grazing (Commonly *Echinochloa crussgalli*, *Echinochloa colonum*, *Cynodon dactylon*, *Elusine indica*, *Digitaria sanguinalis*, *Setaria glauca*, *Azonopus compressus* etc.) from early morning up to afternoon (8 to 10 h) and then returning home. Farmers were supplied a small amount of mixed concentrate (About 30% rice polis, 30% wheat bran, 20% broken rice, and 20% oil cake), 7 to 10 kg per animal per day in the morning regularly. It was also advised to feed their buffalo heifer/cows a sufficient amount of concentrate and green grass supplemented with vitamins and minerals during the breeding season for maintaining good reproductive health. Finally, AI was performed on a total of 160 buffalo cows after observing the estrus sign. The reproductive health of animals was confirmed using manual palpation of the uterus and ovary per rectum.

#### 2.2.3. Grouping of Animals

The grouping of animals was conducted according to the category of the different parameters considering the research hypothesis. The age was determined as previously described by Banerjee et al. [9] and grouped into the following categories: ≤3 to 4, 4.1 to 5, 5.1 to 6, and 6.1 to ≥7 years old. According to the number of previous calving history or parity, it was categorized as parity 0 (not yet calved/heifer), parity 1, 2, 3, and ≥ 4. The body weight was measured by Rondo weighing tape [10]. According to the body weight, animals were classified as ≤200 to 300, 301 to 400, and >400 kg. The reproductive health was scaled as good, moderate, and poor, as previously described by Biswas et al. [4]. The presence or absence of a previous history of calving difficulties was also considered as an influencing factor in pregnancy rate. A vasectomized buffalo bull was used as a teaser bull in this study for estrus detection as well as the farmer’s detection. Therefore, the estrus detection was categorized as estrus detection by farmers and by teaser bulls.

#### 2.2.4. Use of mPLD/Artificial Penis

The penis-like device was used as described by Biswas et al. [4]. Briefly, before inserting the device into the vagina, it was sterilized with 70% ethanol spray and lubricated with coconut oil. After insertion into the vagina, it was pushed and pulled three to four times slowly for stimulation. The device was modified to fit the size and shape of female buffaloes. During AI, it was cleaned and lubricated with gel, and warm water (45–48 °C) was pushed by a 50 mL syringe with an 18 G needle. The AI gun was passed through the device after thawing and loading the AI gun with a semen straw in the way of standard procedure. The biological response was measured in this study by the rate of pregnancy of female buffaloes.

#### 2.2.5. Experimental Design

The study was designed as four experimental groups. Each group consisted of 40 animals homogenously. The experimental intervention was described as follows:

Group A (Control): In this group, buffalo cows/heifers were inseminated by AI technicians after observing heat signs without applying intra-vaginal bio-stimulation.Group B (mPLD): In this group, AI was conducted after observing estrus signs, and intra-vaginal bio-stimulation was applied with a mPLD following artificial insemination to trigger ovulation and increase sperm swimming.Group C (CM): In this group, AI was conducted after observing estrus signs, and a clitoral massage (CM) was applied for 30 s.Group D (mPLD+CM): In this group, both the mPLD and the CM were applied following AI.

#### 2.2.6. Estrus Detection and Insemination

The estrus detection and AI technique were performed as described by Praveen et al. [11]. In this study, the estrus of buffalo cows was detected mostly at night and somewhat in the morning by observing signs of estrus such as drooling of vaginal mucus and standing to be mounted. It was also noticed that excessive bellowing, vulval swelling, restlessness, and temporary teat engorgement. Sometimes, rectal palpation was performed, and a coiled and tonous uterus was indicated as an estrus sign in case of silent heat. Congested vulva and clear mucus streaming were also considered as an estrus sign. We also used a teaser bull for estrus detection in a *bathan* to compare with the estrus detection by farmers. After the determination of estrus, the cow was inseminated by a skilled AI technician within 12 to 18 h of observing estrus household buffaloes. The am–pm method was used for the animals that were reared as *bathan* practice. Single insemination per female buffaloes was considered for this study.

#### 2.2.7. Pregnancy Diagnosis

The absence of estrus signs following 20–25 days of AI was primarily considered as pregnant. The confirmatory diagnosis was conducted using rectal palpation of the reproductive organ between 60 and 90 days post-insemination. 

#### 2.2.8. Statistical Analysis

The study was conducted in field conditions. The collected data were recorded and coded in an Excel sheet. The rate was expressed as a percentage (%). The experiment was designed for between-subjects experimental design with multiple treatment groups to assess the effects of different interventions on AI outcomes. The analysis of variance was calculated by SPSS statistical software (version 20.0). The chi-square test was conducted. The logistic regression was also conducted to analyze the pregnancy-associated risk factors. The test was considered significant at a level of *p* < 0.01 and *p* < 0.05. The data were decoded, entered, and sorted accordingly using MS Excel. The data were then transferred to the SPSS software (Version 25.0 for descriptive analysis. Initially, the data were sorted and cross-checked for duplication and/or missing values. The missing values for each variable were excluded from the analysis. The statistical model was as follows:$$\mathrm{Logit}\{P(Y=1)\}=\mathrm{In}\{\frac{P\left(Y=1\right)}{1-P\left(Y=1\right)}\}=\beta_{0}+\beta_{1}X_{1}+\beta_{2}X_{2}+\ldots +\beta_{n}X_{n}$$
where

P(Y = 1)P(Y = 1)P(Y = 1) = Probability of the success event (e.g., conception or pregnancy).P(Y = 0)P(Y = 0)P(Y = 0) = Probability of failure event (e.g., no conception or pregnancy).β_0_ = Intercept (constant).β_1_, β_2_, …, βn = Coefficients of the predictors.X_1_, X_2_, …, Xn= Independent variables

## 3. Results

### 3.1. Pregnancy Rate in Different Interventions

The bar diagram illustrates the pregnancy rates across four experimental groups A (Control), B (mPLD), C (CM), and D (mPLD+CM), which are indicated in Figure 2. Group D had the highest pregnancy rate of all the other groups. The use of mPLD and clitoral massage subsequent to AI has been shown as an effective protocol that had a higher probability of female buffaloes becoming pregnant. The single use of a mPLD (group B) during AI in buffalo cows also showed a relatively strong performance, though it was approximately 10% lower than that of group D (mPLD+CM). These gaps may suggest that more sexual bio-stimulation during AI is a more favorable condition for achieving the highest chances of pregnancy in buffalo cows.

### 3.2. Factors Influencing the Pregnancy Rate

The results from Table 1 highlight various factors influencing the pregnancy rate in relation to the different experimental groups. The findings indicated the complex interplay of physiological, managemental, and external factors affecting reproductive success.

#### 3.2.1. Age of ANIMALS

The pregnancy rates in ≤3 to 4, 4.1 to 5, 5.1 to 6, and 6.1 to ≥7 years old buffalo cows were 43.75, 45.24, 41.38, and 32.14%, respectively (Table 1). The highest pregnancy rate (45.24%) was observed in animals aged between 4.1 and 5 years old. In this study, there were no significant (*p* = 0.819) differences nor relationships (ꭓ^2^ = 0.925) among these age groups. It was also found that the pregnancy rate in group D was numerically higher than that of other groups.

#### 3.2.2. Parity of Cows

The pregnancy rates in parity 0 (heifer), 1, 2, 3, and ≥4 were 47.22, 40.63, 37.93, 50.00, and 33.33%, respectively (Table 1). There was no significant (*p* = 0.804) variation among the parity groups. However, the statistical data showed a positive relationship (ꭓ^2^ = 1.628). The highest pregnancy rate was found in parity 2 in group D (mPLD+CM).

#### 3.2.3. Body Weight

According to the body weight of buffalo cows, the overall pregnancy rate of ≤200 to 300, 301 to 400, and >400 kg body weight of cows were 33.33, 45.78, and 41.38%, respectively (Table 1). There was no significant difference (*p* = 0.583), but there was a positive relationship (ꭓ^2^ = 1.080) among the groups.

#### 3.2.4. Reproductive Health

Based on reproductive health status (condition of uterus and ovary) during the AI, the overall pregnancy rate in good, moderate, and poor scaling were 35.48, 43.90 and 16.67%, respectively. There was no significant (*p* = 0.320) difference among these criteria. However, the moderate reproductive health status of cows had a more than 2.2 times stronger relationship (ꭓ^2^ = 2.279) than that of poor reproductive health. The pregnancy rates in groups B (mPLD) and D (mPLD+CM) were comparatively higher than those of other groups in connection with the moderate condition.

#### 3.2.5. Calving Difficulties

The cows with a history of previous calving difficulties showed poorer (20%) chances of a high pregnancy rate than those with previous safe and easy parturition (41.94%). 

#### 3.2.6. Heat Detection Methods

The pregnancy rate was higher (61.54%) in the cows that were detected estrus by teaser bulls than that of estrus detected by farmers (39.46%). It was determined that there were no significant (*p* = 0.121) differences; however, it had a more than twice-as-high relationship (ꭓ^2^ = 2.403). In the case of logistic regression analysis, the overall pregnancy rate in the group where estrus signs were detected by the farmer had a significantly (*p* = 0.047) four times higher chance of pregnancy than that of teaser bull’s estrus sign detection (Table 2).

## 4. Discussion

The overall pregnancy rate was found to be 41.3% (66/160) in this study. It is lower than that of Sarker et al. [8] in the coastal region, who found the overall pregnancy rate was 46.4%. This is due to the high pregnancy rate in natural services, which was also included in the average pregnancy rate. The overall rate of pregnancy in the study is similar to the report published by Yousuf et al. [12] and higher than the report of Hoque et al. [13], which were 41.3% and 28.0%, respectively.

According to the experimental protocol, the combined application of intra-vaginal bio-stimulation with a mPLD and clitoral massage after AI (group D) was determined to result in a significantly (*p* < 0.05) higher pregnancy rate than that of the other three experimental groups (A, B, C). The single use of a mPLD (group B) also increased the pregnancy rate more than that of the control group (group A) and the single use of clitoral massage (group C). In group B, we used a mPLD (artificial penis) that gave bio-stimulation and hastened the ovulation process, which increased the pregnancy rate by more than only the AI service group. In this study, the experimental groups were not compared with the natural service because our previous study [8] was designed with three experimental groups that were natural service, only AI, and AI with the use of PLD groups, in which the pregnancy rate was 64, 32, and 44%, respectively. However, the PLD was used similarly to the design for cattle [4], which was modified in this study. Therefore, the natural service was not considered in this experiment. The buffalo cows were approachable to intra-vaginal bio-stimulation because the mPLD may help them to have feelings of a bull’s penis. Therefore, a relatively higher pregnancy rate was found [8]. Although proper feeding and nutrient management directly influence reproductive performance, affecting estrus cycle regularity, conception rate, and embryonic survival [11], intra-vaginal bio-stimulation helps to achieve feelings of the bull penis, triggering ovulation and stimulating cyclic activity. Consequently, this enhances the pregnancy rate of buffalo heifers/cows [4]. The artificial penis may also influence the secretion of oxytocin hormone (not determined in this study), which significantly triggers sperm transportation and the release of the LH surge. In the study, a balanced diet was supplied to the animals, which may also increase the chances of physiological response to pregnancy rate. A well-balanced diet significantly enhances buffalo fertility and pregnancy rates. Strategic feeding practices focusing on energy balance, essential nutrients, and proper body condition can maximize reproductive efficiency and overall productivity [11]. Previously, it was stated that the genital tract, such as clitoral stimulus at the time of AI, might favorably influence pregnancy rates in cattle, which was shown in several studies to improve pregnancy rates by 6.3 to 7.5% in cows [5,7]. The findings of our study are similar to the results of Sarker et al. [8] in the selected coastal region of Bangladesh. Biswas et al. [4] also reported that PLD stimulation increases the rate of pregnancy in cows during bio-stimulation to the vagina. Bull penis stimulation hastens the reproductive effectiveness, according to Choudhary et al. [14], and follicular development of anoestrus heifers [15]. The side effects of the use of the device could not be noticed during the study; however, the repeated use of the device for a long time could not be determined yet. In our previous study of cattle, no significant effect was noticed by the farmers [4].

The highest pregnancy rate (45.24%) was observed in animals aged between 4.1 and 5 years old, which aligns with previous findings that animals in their prime reproductive age exhibit better fertility [16]. Hamid et al. [17] also reported the highest pregnancy rate (67.8%) among buffalo cows within the 4.5 to 5.6 years group of age. Parity, or the number of previous pregnancies, showed a notable influence on pregnancy rates. Animals with three previous pregnancies (P3) recorded the highest rate (50.00%), which may reflect improved uterine health and better adaptation to calving cycles [18]. In contrast, animals with no previous pregnancies (P0) and higher parity numbers (>P4) exhibited lower pregnancy rates (33.33%), which could indicate challenges such as uterine scarring or hormonal imbalances. Conversely, Sarker et al. [8] found that the highest pregnancy rate in parity two buffalo cows but had no significant differences with parity three cows. Additionally, Bhagat and Gokhale [19] noted a gradual increase in the pregnancy rate from the first to the fourth parity, followed by a decrease in the subsequent parities, which is a similar trend in our study. Body weight played a significant role in the pregnancy rate, with animals weighing between 301 and 400 kg achieving the highest rate (45.78%). Underweight animals (≤200 to 300 kg) had the lowest rates (33.33%), suggesting a link between optimal body condition and reproductive success. These findings corroborate the work of Brown et al. [20], who emphasized the importance of maintaining ideal body condition for reproductive performance. Reproductive health (RH) was a critical determinant, with animals categorized as having “good” reproductive health showing a pregnancy rate of 43.90%. Poor reproductive health resulted in a starkly lower rate of 16.67%, underlining the need for effective health management protocols to mitigate reproductive disorders [21]. Similarly, Sarker et al. [8] reported that The buffaloes with moderate RH were shown significantly (*p* < 0.05) higher pregnancy rate (79.16%) than those of poor and good scale. Mufti et al. [22] also reported that reproductive disorders in the remaining 15% of the heifer/cows were an important cause of reduced pregnancy rates, which was partially supported by this study. Animals with no calving difficulties recorded a substantially higher pregnancy rate (37.94%) compared to those with prior calving difficulties (20.00%). This finding aligns with earlier studies that highlight the adverse impact of dystocia on subsequent fertility [23]. Simultaneously, Sarker et al. [8] also determined that the significantly (*p* < 0.001) highest pregnancy rate (54.68%) was found in the case of the absence of calving difficulties. Our observations align with those of Sarker et al. [8] (2022), who reported that the presence of a history of previous calving difficulties negatively influences reproductive usefulness and pregnancy rates in bovines. The method of heat detection significantly influenced outcomes, with the use of teaser bulls resulting in a higher pregnancy rate (61.54%) compared to detection by farmers (39.46%). This aligns with the findings of Garcia et al. [24] and Paul et al. [25], emphasizing the accuracy and reliability of biological detection methods. Across the groups, there were clear variations in pregnancy rates, reflecting the cumulative effect of these factors. Group D consistently showed superior performance across several categories, suggesting the presence of more effective interventions or management practices in this group. The study underscores the multifactorial nature of fertility in animals and the importance of targeted management strategies. Special attention should be given to maintaining optimal body condition, ensuring reproductive health, and employing reliable heat detection methods to improve reproductive outcomes.

There are some limitations in the study. The animals in the study were in the same management system at the field level, and there was an absence of close monitoring of the intensive farming system. The animals were reared in the *bathan* practice. The experimental data were achieved in field conditions. Notably, the hormonal assay was not conducted in this study. It is important that future studies can be supported by a reproductive and metabolic hormonal assay to determine the mechanism pathway in the pregnancy rate response.

## 5. Conclusions

It is concluded that the application of a mPLD in conjunction with massage of the clitoris greatly increased the pregnancy rate of buffalo cows. The study lacks results that can be supported by a reproductive or metabolic hormonal assay to determine the pathway mechanism in the pregnancy rate response.

## Figures and Tables

**Figure 1 animals-15-00597-f001:**
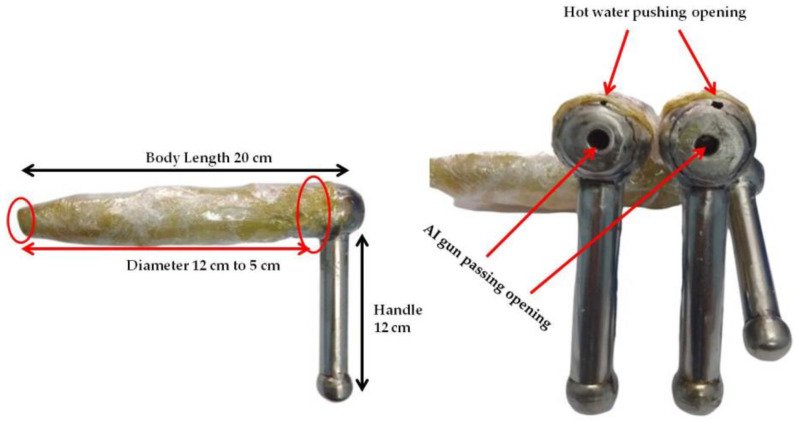
Artificial bull penis or modified penis-like device (mPLD).

**Figure 2 animals-15-00597-f002:**
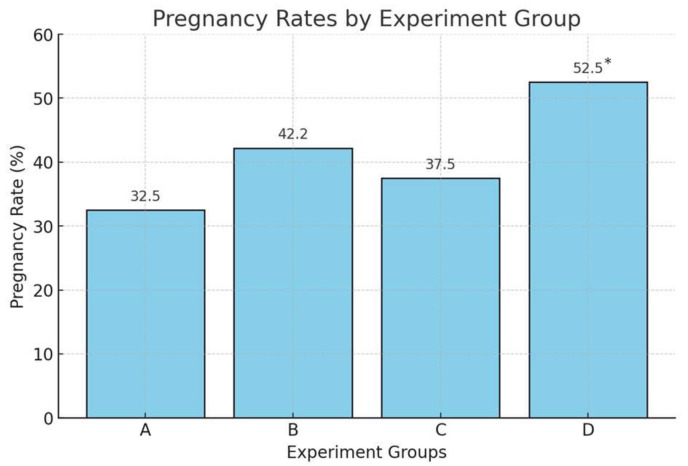
Pregnancy rate in different interventions (A: Control, B: mPLD, C: CM, D: mPLD+CM; * indicates *p* < 0.05).

**Table 1 animals-15-00597-t001:** Factors influencing pregnancy rates in female buffaloes in four experimental groups.

Factors	Category (n)	Pregnancy Rate % (n)	ꭓ^2^ & *p* Value	Group Wise Pregnancy Rate n (%)			
				A	B	C	D
Age (year)	≤3 to 4 (32)	43.75 (14)	0.925 0.819	0 (0.0)	5 (21.43)	5 (35.71)	6 (42.86)
	4.1 to 5 (42)	45.24 (19)		2 (10.53)	5 (26.32)	3 (15.79)	9 (47.37)
	5.1 to 6 (58)	41.38 (24)		6 (25.00)	7 (29.17)	6 (25.00)	5 (20.83)
	>6 to ≥7 (28)	32.14 (9)		5 (55.56)	2 (22.22)	1 (11.11)	1 (11.11)
Parity (number)	P0 (36)	47.22 (17)	1.628 0.804	3 (17.65)	10 (58.82)	2 (11.76)	2 (11.76)
	P1 (32)	40.63 (13)		3 (23.08)	3 (23.08)	4 (30.77)	3 (23.08)
	P2 (58)	37.93 (22)		4 (18.18)	2 (9.09)	5 (22.73)	11 (50.00)
	P3 (24)	50.00 (12)		3 (25.00)	1 (8,33)	3 (25.00)	5 (41.67)
	≥P4 (10)	33.33 (2)		0 (0.0)	1 (50.00)	1 (50.00)	0 (0.0)
Body weight (kg)	≤200 to 300 (48)	33.33 (16)	1.080 0.583	3 (18.75)	4 (25.00)	2 (12.50)	7 (43.75)
	301 to 400 (83)	45.78 (38)		8 (21.05)	11 (28.95)	8 (21.05)	11 (28.95)
	>400 (29)	41.38 (12)		2 (16.67)	2 (16.67)	5 (41.67)	3 (25.00)
Reproductive health (RH)	Good (31)	35.48 (11)	2.279 0.320	3 (27.27)	2 (18.18)	5 (45.45)	1 (9.09)
	Moderate (123)	43.90 (54)		10 (18.52)	14 (25.93)	10 (18.52)	20 (37.04)
	Poor (6)	16.67 (1)		0 (0.0)	1 (100.0)	0 (0.0)	0 (0.0)
Calving difficulties (CD)	Yes (5)	20.00 (1)	0.962 0.327	0 (0.0)	0 (0.0)	1 (100.0)	0 (0.0)
	No (155)	41.94 (65)		13 (20.00)	17 (26.15)	14 (21.54)	21 (32.31)
Heat detection	By farmer (147)	39.46 (58)	2.403 0.121	10 (17.24)	14 (24.14)	14 (24.14)	20 (34.48)
	By teaser bull (13)	61.54 (8)		3 (37.50)	3 (37.50)	1 (12.50)	1 (12.50)

Indicated A: Control; B: mPLD; C: CM; D: mPLD+CM.

**Table 2 animals-15-00597-t002:** Analysis of influencing factors for buffalo cow pregnancy rates.

Factors	Variables	Coefficient	Standard Error	Wald	Sig.	Odd Ratio	95% Confidence Interval for Odd Ratio	
							Lower Bound	Upper Bound
Age	≤3 to 4	-	-	1.648	0.649	-	-	-
	4.1 to 5	−0.271	0.520	0.271	0.603	0.763	0.309	2.626
	5.1 to 6	−0.331	0.464	0.510	0.475	0.718	0.323	2.104
	>6 to ≥7	0.301	0.530	0.322	0.571	1.351	0.488	4.145
Parity	P0	-	-	3.028	0.553	-	-	-
	P1	−1.110	0.977	1.292	0.256	0.330	0.050	2.306
	P2	−0.718	1.006	0.510	0.475	0.488	0.071	3.757
	P3	−0.435	0.938	0.215	0.643	0.647	0.127	5.458
	P4	−0.950	0.995	0.912	0.340	0.387	0.063	3.367
Body weight	≤200 to 300	-	-	2.195	0.334		-	-
	301 to 400	0.342	0.560	0.373	0.541	1.407	0.487	4.894
	>400	−0.280	0.514	0.297	0.586	0.756	0.282	2.238
Reproductive health	Good	-	-	2.551	0.279		.	.
	Moderate	−1.087	1.249	0.757	0.384	0.337	0.032	4.533
	Poor	−1.531	1.154	1.761	0.184	0.216	0.028	2.684
Calving difficulties	Yes	-	.	.	.	.	.	.
	No	1.482	1.186	1.561	0.212	4.400	0.390	42.267
Heat detection	By farmer	-	.	.	.	.	.	.
	By teaser bull	1.394	0.701	3.951	0.047	4.032	1.109	18.798

## Data Availability

The original contributions presented in this study are included in the article. Further inquiries can be directed to the corresponding author.

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
