# Peer review of "Intra-Vaginal Bio-Stimulation and Clitoral Massage to Enhance Pregnancy Rates in Water Buffalo in Coastal Bangladesh†"

_animals, 2025, doi:10.3390/ani15040597_

Round 1

Reviewer 1 Report

Comments and Suggestions for Authors

General Evaluation:

In the manuscript, the authors examined the effect of an artificial penis (modified penis-like device - mPLD) and vaginal biostimulation with clitoral massage to increase the pregnancy rate in water buffalos. The subject of this study is suitable for the “Animals” journal. The article focuses on a unique and important topic because the reproductive efficiency of water buffalos is of great economic importance in developing countries like Bangladesh. The study included four experimental groups where animals were divided into different groups, and different biostimulation techniques were used. The findings show that artificial penis and clitoral massage were effective in increasing pregnancy rate. The study is well designed and presented, but I offer a few corrections to increase the scientific value of the manuscript. After the authors address these corrections, the manuscript can be accepted.

1. Title and Abstract:

The title accurately reflects the content of the article. The purpose of the study and the methods used are clearly stated. However, the title and abstract may be a bit long and may bore the general reader. The title could be shorter. The abstract adequately explains the purpose of the study, the methods used, and the results obtained. However, some parts could be more clearly defined, mainly how biostimulation was applied and measured.

2. Introduction:

The introduction provides a general perspective on why the study is essential and its place in the literature. In particular, information on the economic and ecological role of water buffalos in Bangladesh is well-provided. However, some additions could be made. For example, a summary of other previous studies in this field and the findings obtained could be provided, thus contributing to the study to the literature more apparent. In addition, more explanations could be given about the differences between biostimulation and artificial insemination. However, more details could be given about how interventions such as mPLD (artificial penis) and clitoral massage worked.

3. Methods:

The study design and methods are explained in great detail. The formation of experimental groups, animal selection, devices used, and biostimulation methods applied are well described. However, some improvements can be made in this section as follows:

Animal Selection: Grouping buffalo cows according to age and body weight is suitable. However, more information can be provided about possible biases and limitations in this selection.

Device Description: How the artificial penis (mPLD) and clitoral massager are applied in detail is explained. However, more information can be provided about what biological response the design of the mPLD device targets and how this response is measured.

Statistical Analysis: Statistical analyses appear to have been performed correctly, but some critical points could be explained more clearly. In particular, a more in-depth discussion of the differences between groups and logistic regression analyses would be helpful.

4. Results:

In the results section, it is clearly stated that the interventions performed affect pregnancy rates. It is especially emphasized that the pregnancy rate in group D (mPLD and clitoral massage) is higher than in the other groups. This result is an essential finding that biostimulation can increase pregnancy rates. However, whether the differences between the groups are statistically significant or not can be discussed more clearly. In addition, comments can be made about potential side effects or complications. For example, more information can be provided about the long-term effects of these biostimulation applications and their effects on animal health.

5. Discussion:

In the discussion section, the findings obtained are evaluated by comparing them with the literature. In this section, the comparisons made with previous studies are especially valuable. However, some additions can be made to strengthen this section by authors' suggestions, such as;

Control Group: There are only four groups in the study, but the results can be made more comprehensive by using more control groups (e.g., only natural insemination or only artificial penis).

Biological Mechanisms: The mechanism by which biostimulation increases pregnancy rates can be discussed in more detail. More information on hormone levels and physiological changes could be provided.

Limitations: The limitations of the study could be made more apparent. For example, the different living conditions, health status, and past reproductive success of the animals could affect the results.

6. General Assessment and Recommendations:

The article provides essential findings on the reproductive efficiency of water buffalos and demonstrates the effectiveness of biostimulation techniques. However, there are several areas where improvements could be made:

Biostimulation Mechanism: The biological effects of mPLD and clitoral massage should be discussed in more detail.

Control Groups and Side Effects: More control groups and information on side effects could be provided.

Long-Term Effects: Information on the long-term effects of these biostimulation methods could be added.

In conclusion, the study provides essential findings in a valuable area, but some methodological and discussion sections may require more depth and explanation.

Author Response

Comments 1: Title and Abstract: The title accurately reflects the content of the article. The purpose of the study and the methods used are clearly stated. However, the title and abstract may be a bit long and may bore the general reader. The title could be shorter. The abstract adequately explains the purpose of the study, the methods used, and the results obtained. However, some parts could be more clearly defined, mainly how biostimulation was applied and measured.

Response 1: Thank you for pointing this out. I/We agree with this comment. Therefore, I/we have concise the title as “ Intra-vaginal Bio-stimulation and Clitoral Massage to Enhance Pregnancy Rates in Water Buffalo in Coastal Bangladesh’ . The abstract has been adjust within199 words.

Comments 2: Introduction: The introduction provides a general perspective on why the study is essential and its place in the literature. In particular, information on the economic and ecological role of water buffalos in Bangladesh is well-provided. However, some additions could be made. For example, a summary of other previous studies in this field and the findings obtained could be provided, thus contributing to the study to the literature more apparent. In addition, more explanations could be given about the differences between biostimulation and artificial insemination. However, more details could be given about how interventions such as mPLD (artificial penis) and clitoral massage worked.

Response 2: Agree. I/We have, accordingly, added some more citation relevently to emphasize this point. The added sentences are highlighted green in the text. That are “It is reported that the pregnancy rate in buffalo in the coastal region of Bangladesh in natural service was 64%, whereas it was 32% in AI [8]. Previously, a penis-like device (PLD) was used after AI for intra-vaginal sensation and got about 15.5% higher pregnancy rates in cow [4] and also 12% higher in female buffaloes [8] when compared with only AI technique.”

Comments 3. Methods:

The study design and methods are explained in great detail. The formation of experimental groups, animal selection, devices used, and biostimulation methods applied are well described. However, some improvements can be made in this section as follows:

Comments 3.1. Animal Selection: Grouping buffalo cows according to age and body weight is suitable. However, more information can be provided about possible biases and limitations in this selection.

Response 3.1: Thank you for your suggestions. I have revised some points which is highlighted in green color in the text.

Comments 3.2. Device Description: How the artificial penis (mPLD) and clitoral massager are applied in detail is explained. However, more information can be provided about what biological response the design of the mPLD device targets and how this response is measured.

Response 3.2: Agreed with you. I have inserted some sentences about the use of the device. The biologoical response only meausred by the rate of conception/pregnancy of animals, not by hormonal or metabolic assays. These limitation has been mentioned in the limitation of study paragraph. The sentenceThe biological response was measured in this study by the rate of pregnancy of female buffaloes.” has beed added in the paragraph.

Comments 3.3. Statistical Analysis: Statistical analyses appear to have been performed correctly, but some critical points could be explained more clearly. In particular, a more in-depth discussion of the differences between groups and logistic regression analyses would be helpful.

Response 3.3:  Thank you for your suggestions. I have been revised some points in the discussion to make it more clear.

Comments 4. Results:

In the results section, it is clearly stated that the interventions performed affect pregnancy rates. It is especially emphasized that the pregnancy rate in group D (mPLD and clitoral massage) is higher than in the other groups. This result is an essential finding that biostimulation can increase pregnancy rates. However, whether the differences between the groups are statistically significant or not can be discussed more clearly. In addition, comments can be made about potential side effects or complications. For example, more information can be provided about the long-term effects of these biostimulation applications and their effects on animal health.

Response 4. It is the important point that you identified. I have revised the graph (Figure 2) with indicating the significance difference level by * marks. I hope it is now easily understandable. I have added some sentences about the side effect of long time use in the discussion section. The following sentences has been included The side effects of the use of the device could not be noticed during the study, however, the repeated use of the device for long time could not be determined yet. In our previous study of cattle, there is no significant effect was noticed by the farmers [4].”

Comments 5. Discussion:

In the discussion section, the findings obtained are evaluated by comparing them with the literature. In this section, the comparisons made with previous studies are especially valuable. However, some additions can be made to strengthen this section by authors' suggestions, such as;

Comments 5.1: Control Group: There are only four groups in the study, but the results can be made more comprehensive by using more control groups (e.g., only natural insemination or only artificial penis).

Response 5.1:  It is remarkable points. I have added the following explanation in the discussion section. “In this study, the experimental groups were not compared with the natural service, because in our previous study [8] was designed with three experimental groups that were natural service, only AI and AI with the use of PLD groups, in which the pregnancy rate was 64, 32 and 44%, respectively. However, the PLD was used as similar as designed for cattle [4], which has been modified in this study. Therefore, the natural service was not considered in this experiment.”

Comments 5.2: Biological Mechanisms: The mechanism by which biostimulation increases pregnancy rates can be discussed in more detail. More information on hormone levels and physiological changes could be provided.

Response 5.2:  Thank you. I have mentioned this limitation at the end of discussion. We could not assay the hormone level in this study.

Comments 5.3: Limitations: The limitations of the study could be made more apparent. For example, the different living conditions, health status, and past reproductive success of the animals could affect the results.

Response 5.3:  I have revised the limitations accordingly.

Comments 6. General Assessment and Recommendations:

The article provides essential findings on the reproductive efficiency of water buffalos and demonstrates the effectiveness of biostimulation techniques. However, there are several areas where improvements could be made:

Biostimulation Mechanism: The biological effects of mPLD and clitoral massage should be discussed in more detail.

Control Groups and Side Effects: More control groups and information on side effects could be provided.

Long-Term Effects: Information on the long-term effects of these biostimulation methods could be added.

In conclusion, the study provides essential findings in a valuable area, but some methodological and discussion sections may require more depth and explanation.

Response:  Thank you so much for summarized comments. I have revised and clarified above-mentioned points in the text accordingly your suggestions and comments.

Reviewer 2 Report

Comments and Suggestions for Authors

The manuscript “Effect of Intra-vaginal Bio-stimulation with artificial penis and Clitorial Massage to Increase the Pregnancy Rate of Water Buffalo in the Coastal Region of Bangladesh” has some serious flaws that need to be corrected before it can be considered for publishing.

First, it is unclear if the groups were homogeneous or how the 40 animals in each group were allocated. If they were not homogeneous, how do you guarantee the results were due to the treatment? It is also advisable to change the group names from A, B, C, and D to something that hints at what was done to help the reader remember which treatment you are talking about instead of repeating it over and over throughout the manuscript. 

The second most serious flaw is the analysis performed. The data should be analyzed as the interaction between groups and the different factors. According to this, the results and discussion need to be restructured. 

Also, a complete review of written English is recommended as there are some grammar mistakes throughout the manuscript. 

Comments on the Quality of English Language

The English language must be improved, as there are several grammatical mistakes. Some sentences are unclear.

Author Response

1. Summary

I would like express my gratitude for taking the time to review this manuscript. Please find the detailed responses below and the corresponding revisions/corrections highlighted in the re-submitted files.

2. Questions for General Evaluation

Reviewer’s Evaluation

Response and Revisions

Does the introduction provide sufficient background and include all relevant references?

Must be improved

It has been checked and improved accordingly. The English language has been rechecked and corrected.

Are all the cited references relevant to the research?

Must be improved

Is the research design appropriate?

Must be improved

Are the methods adequately described?

Must be improved

Are the results clearly presented?

Must be improved

Are the conclusions supported by the results?

Must be improved

3. Point-by-point response to Comments and Suggestions for Authors

Comments 1: The manuscript “Effect of Intra-vaginal Bio-stimulation with artificial penis and Clitorial Massage to Increase the Pregnancy Rate of Water Buffalo in the Coastal Region of Bangladesh” has some serious flaws that need to be corrected before it can be considered for publishing.

First, it is unclear if the groups were homogeneous or how the 40 animals in each group were allocated. If they were not homogeneous, how do you guarantee the results were due to the treatment? It is also advisable to change the group names from A, B, C, and D to something that hints at what was done to help the reader remember which treatment you are talking about instead of repeating it over and over throughout the manuscript. 

Response 1: Thank you for pointing out important issues of this study. During the selection of animals, we selected about 250 female buffaloes for the experiment, and provided and suggested a similar management system as mentioned in the methodology section. Among them, we have categorized homogeneously 40 animals for each treatment group of the experiment. However, not all parameters/factors could meet the same numbers for analyzing the determination of influencing factors. It is also indicated that a single artificial insemination per female buffalo was considered for this study. I have included a sentence in the 2.2.6 paragraph which is “There was single insemination per female buffaloes was considered for this study.” In the case of experimental group names, I have mentioned the A, B, C, and D indications in the figure title of the graph.

Comments 2: The second most serious flaw is the analysis performed. The data should be analyzed as the interaction between groups and the different factors. According to this, the results and discussion need to be restructured. 

Response 2:  Thank you so much for your comments. I have shown the interaction between groups and different factors in Table 1. For better understanding, I have revised the table title as “Factors influencing pregnancy rate in female buffaloes in four experimental groups”

Comments 3:  Also, a complete review of written English is recommended as there are some grammar mistakes throughout the manuscript. 

Response 2:  I am so sorry for the grammar mistakes. I have checked and corrected accordingly. 

Reviewer 3 Report

Comments and Suggestions for Authors

Comments and suggestions for the authors are indicated in the attached document written in Word.

Author Response

Response to Reviewer 3 Comments

1. Summary

Thank you very much for taking the time to review this manuscript. Please find the detailed responses below and the corresponding revisions/corrections highlighted with green text color in the re-submitted files.

2. Questions for General Evaluation

Reviewer’s Evaluation

Response and Revisions

Does the introduction provide sufficient background and include all relevant references?

Can be improved

It is has been improved as suggested.

Are all the cited references relevant to the research?

Can be improved

Is the research design appropriate?

Can be improved

Are the methods adequately described?

Can be improved

Are the results clearly presented?

Can be improved

Are the conclusions supported by the results?

Can be improved

3. Point-by-point response to Comments and Suggestions for Authors

TITLE

25 words, meets the indication: <20 words.

Review why some words are not capitalized.

Response

“Intra-vaginal Bio-stimulation and Clitoral Massage to Enhance Pregnancy Rates in Water Buffalo in Coastal Bangladesh”

SIMPLE

SUMMARY

149 words, <150 words.

Page 1, line 24, remove the repeating point.

Response

Now it is 149 words. I have added line “The penile sensation during mating is important for successful fertilization.”

The point has been rermoved.

ABSTRACT

211 words; in general, review the journal's guidelines for writing the abstract of a scientific article.

Page 1, line 32; it is suggested to write: the buffalo cows were divided into four experimental groups (group A…

Page 1, lines 41-42, it is suggested to write: conjugation with

massage of the clitoris enhances the pregnancy rate of buffalo cows in the Coastal Region of Bangladesh.

Response

Now it is 199 words. Removed line is “The study was conducted from July 2023 to June 2024 in the selected coastal region of Bangladesh.”

Both sentence has been revised it according to your suggestions.

KEYWORDS

3 words, meets the indications: <5 words.

These keywords are repeated in the title and will not give the opportunity to expand the search for information. It is suggested not to repeat simple or compound words that are already included in the title. It is suggested to write other keywords,  such  as:  Bubalus  bubalis,  reproductive

management, coastal livestock farming.

Response

“Arificial penis, Bio-stimulation, Pregnancy rate’ are replaced with “Bubalus  bubalis,  reproductive management, coastal livestock farming”

INTRODUCTION

Check that this section presents literature from the last ten years.

Page 2, line 45, write Bubalus bubalis in italics.

Page 2, lines 55-57, it is suggested to write: farmers alike. Low availability of feed or low quality of forage required during reproductive events, inadequate estrus detection and artificial insemination (AI) efficiency often limit successful breeding in buffalo.

Page 2, lines 87-88, write: Therefore, the objective of this study

is…

Response

It has made in italic.

I have replaced this line “Inadequate estrus detection and artificial insemination (AI) efficiency often limit successful breeding in buffalo.” With your suggested sentence.

It has been revised.

MATERIAL AND METHODS

The information in this section must be consistent with the objective of the study.

Page 3, line 97, indicate whether the study was conducted with ethical approval from any Animal Ethics Committee of the University.

Response: I have mentioned in the text now.

Page 3, lines 101-103; separate numbers from units of measurement and write the symbols correctly: levamisole 600 mg, triclabendazole 900 mg, 2 g, 75 kg, 30 mL, 10mL

Page 3, lines 110-111, separate numbers from units of measurement: 20 cm, 12 cm, 5 cm, 12 cm; as can be seen in the text inside Figure 1, which describes the dimensions of artificial bull penis.

Response: It has been corrected accordingly.

1

Page 3, line 113, it is suggested to write: … handle to push warm water into the device for a warm sensation like an erect penis buffalo bull.

Response: It has been corrected accordingly.

Page 3, line 116, to write: Figure 1: Artificial Bull penis or modified penis like device (mPLD)

Response: It has been corrected accordingly.

Page 4, line 142, describe the methodology of estrus detection in female buffaloes, or insert a citation referring to the methodology.

Response: Thank you. It has been described in ‘2.2.6. Estrus detection and insemination’ with reference.

Page 4, line 143, write the scientific name of the forage species where the female buffaloes grazed and indicate the time spent (in hours) on the pasture.

Response: It has been included accordingly.

Page 4, lines 145-146, describe the percentage of ingredients included in the mixed concentrate and indicate the grams of supplement offered to female buffaloes by farmers per day. Mention the times the supplement was offered per day and the duration of the supplementation period.

Response: It has been indicated accordingly.

Page 4, lines 146-147, explain whether the supplementation was offered before, during or after the detection of estrus; and explain how the supplementation was carried out.

Response: It has been corrected and clarified accordingly in the text.

Page 4, lines 148-149, describe the AI methodology in female buffaloes, or insert a citation that refers to the methodology.

Response: A reference has been included.

Page 3, line 159, separate numbers from units of measurement: >400 kg

Response: It has been corrected accordingly.

Page 3, lines 187-196, the literature reports that in buffaloes most estruses occur at night. Explain whether in the present study estrus detection was performed at night, in the morning or throughout the day.

Response: It is mostly at night and few are in the morning. I have added a sentence..

Page 5, lines 216-224, were the data analyzed with a normality test to consider the use of parametric statistics?

Response: Yes, It has been tested by paradigm model. The data were homogenously distributed in each group.

Mainly with the data expressed in percentage units. In addition, the data conditions must present equal variances and independence.

Mention the experimental design and statistical model used.

Response: It has been included accordingly.

RESULTS

Page 5, line 217, it is possible that the percentage values are indicated only in the Figure 2 and not repeated in the text, so as not to cause redundancy.

Response: It has been corrected accordingly.

Page 5, lines 217-218, at what level of significance was the pregnancy rate higher in experimental group D; 0.05 or 0.01? Page 6, lines 233-238, while Table 1 describes several factors influencing pregnancy rate in relation to the different experimental groups, the P values for each factor were not significant at any level (0.05 or 0.01) for pregnancy rate by group in the present study. Therefore, the title of Table 1 could be rephrased as: Table 1. Factors influencing pregnancy rate in female buffaloes in four experimental groups.

Response: It has been corrected and rephrases of table 1 title accordingly.

Page 7, lines 244-245, 250-251, how are these ideas argued? If Table 1 indicates that the P values were not significant?

Response: The p>0.05 which is not significantly different. The chi-square value is shown positive value, which meant there is a relation with each others.

Page    7,   line    255,   separate   numbers   from   units    of

measurement: >400 kg

Response: It has been corrected.

DISCUSSION

Page 8, lines 281-253, In general, the evidence was found to be quite strong and fairly well supported by the references cited.

Although the assumptions based on the results of the statistical analyses are likely to be reconsidered, and it is indicated whether the results of the pregnancy rate were different at a significance level for each factor between the four experimental groups.

Page 8, lines 288-296, In the Results information, described in section 3.1. Pregnancy rate in the different interventions, the level of significance with which the pregnancy rate of experimental group D was compared with that of the other three groups (A, B, C) is not specifically indicated. Although Figure 2 only shows graphically and numerically the percentage values of the pregnancy rate between groups, it is not clear how the significant differences between groups were found.

Therefore, it is possible that the arguments written in these lines (288-296) of the Discussion section do not agree with what is indicated in Figure 2.

Response: It has been corrected. The star mark is added in the figure 2 to indicate the significance variance. The group D was significantly (p<0.05 difference from other group.

It is suggested to improve the description in section 3.1. (Pregnancy rate in the different interventions) and indicate the P values to understand how differences were found between the experimental groups for the pregnancy rate.

Response: It has been improved in the section.

Page 9, lines 299-303, it is possible to speculate about some physiological responses that occur because it is known that this is reported in the literature, but it must be evidenced by the results obtained and by the support of statistical analysis of the data.

For example, something that was mentioned in the methodology but not included in the discussion is the supplementation that the female buffaloes received or the grazing management that was provided to some animals during the experiment. It is possible to infer that during the physiological reproductive processes, the nutrition or feeding factor directly influences the pregnancy rate in female mammals.

Response: It has been improved in the section.

Page 10, lines 354-358, It is undeniable that most studies with experimental animals have some limitations. If the animals in the study were not in the same management system, one can choose to consider alternatives in the experimental design. If most of the animals were raised in the bathan practice and the experimental data were obtained under field conditions, it is acceptable so as not to alter the conditions under which the females express their reproductive response. It is important that future studies can be supported by a reproductive and metabolic hormonal assay to determine the mechanism pathway in the pregnancy rate response.

Response: It has been corrected according to your suggestions. The animals were in same manangement system.

CONCLUSIONS

The conclusion should be consistent with the stated objective. The objective of this study is: to evaluate the effect of intra- vaginal bio-stimulation using a modified penis-like device and clitoral massage on the pregnancy rates of water buffalo in the coastal region of Bangladesh.

Therefore, it is suggested not to assume that GnRH hormone injection after application of the technique can increase the pregnancy rate of water buffaloes, since this study lacks results that can be supported with a reproductive or metabolic hormonal assay to determine the pathway mechanism in the

pregnancy rate response.

Response: It has been rewritten according to your suggestions.

REFERENCES

Check that all citations have their bibliographical references written in this section.

Perhaps the DOI address of all references can be inserted.

Response: It has beedn rechecked and included.

Round 2

Reviewer 1 Report

Comments and Suggestions for Authors

The authors have made all corrections required in the manuscript. The revised version of the article can be acceptable for publication.

Author Response

2nd Response to Reviewer 1

1. Summary

Thank you very much for taking the time to re-review this manuscript.

2. Questions for General Evaluation

Reviewer’s Evaluation

Response and Revisions

Does the introduction provide sufficient background and include all relevant references?

Yes

It is my pleasure for your extended support to improve the manuscript.

Are all the cited references relevant to the research?

Yes

Is the research design appropriate?

Yes

Are the methods adequately described?

Yes

Are the results clearly presented?

Yes

Are the conclusions supported by the results?

Yes

3. Point-by-point response to Comments and Suggestions for Authors

Comments 1: The authors have made all corrections required in the manuscript. The revised version of the article can be acceptable for publication.

Response 1: It is our great pleasure and showing my heartfelt gratitude to you.

Reviewer 2 Report

Comments and Suggestions for Authors

The manuscript has improved since the previous review. Nonetheless, I still recommend that it be clearly stated if the animals were allocated in each experimental group homogeneously or how the 40 animals of each group were chosen for said group. 

Also, the group names need to be changed from A, B, C, and D to something like: control, mPLD, Clit massage (or CM), and mPLD+clit massage (or mPLD+CM). Something that you don't need to review the manuscript to remember what it stands for.

Author Response

2nd Response to Reviewer 2

1. Summary

Thank you very much for taking the time to re-review this manuscript.

2. Questions for General Evaluation

Reviewer’s Evaluation

Response and Revisions

Does the introduction provide sufficient background and include all relevant references?

Yes

It is my pleasure for your extended support to improve the manuscript.

Are all the cited references relevant to the research?

Can be improved

Is the research design appropriate?

Can be improved

Are the methods adequately described?

Yes

Are the results clearly presented?

Yes

Are the conclusions supported by the results?

Yes

3. Point-by-point response to Comments and Suggestions for Authors

Comments 1: The manuscript has improved since the previous review. Nonetheless, I still recommend that it be clearly stated if the animals were allocated in each experimental group homogeneously or how the 40 animals of each group were chosen for said group. 

Response 1: I am so sorry that I could not make you clear understanding about it. In the section of the materials and method about the selection of animals “2.2.2. Selection and management of buffalo heifers/cows A total of 200 buffalo heifers/cows were selected by simple random sample method from the study area for the experiment based on the research communication, health status, calving interval, farmer’s accountability, etc. Different types of inspections were performed to select buffalo heifers/cows. Age, parity, body condition score, reproductive health status, and previous calving difficulties were recorded.”

It is randomly selected. The number of animals (40) in each group were homogenously selected for the study. It is my mistakes that I could not mention in the text previously. Now I have mentioned as “Each group consisted of 40 animals homogenously.”

Comments 2: Also, the group names need to be changed from A, B, C, and D to something like: control, mPLD, Clit massage (or CM), and mPLD+clit massage (or mPLD+CM). Something that you don't need to review the manuscript to remember what it stands for.

Response 2: I am sorry that could not indicate in the manuscript. Now I have indicated in the experimental design, in figure legend, bottom of the table as well as in the text where it is mentioned.

The blue text color is the new changes in the manuscript.

Thank you again for your patient review and wise comments and suggestions.
